# Nonlinear Effects of the Built Environment on Light Physical Activity among Older Adults: The Case of Lanzhou, China

**DOI:** 10.3390/ijerph19148848

**Published:** 2022-07-21

**Authors:** Peng Zang, Hualong Qiu, Fei Xian, Linchuan Yang, Yanan Qiu, Hongxu Guo

**Affiliations:** 1Department of Architecture, Guangdong University of Technology, Guangzhou 510006, China; kenxin8989@163.com (P.Z.); bkj338@163.com (H.Q.); 18308463173@163.com (F.X.); stewart87@sina.com (Y.Q.); 2Department of Urban and Rural Planning, School of Architecture, Southwest Jiaotong University, Chengdu 611756, China; yanglc0125@swjtu.edu.cn

**Keywords:** green visibility, built environment, light physical activity, random forest, older adults

## Abstract

The aging of the population is increasing the load on the healthcare system, and enhancing light physical activity among older adults can alleviate this problem. This study used medical examination data from 1773 older adults in Lanzhou city (China) and adopted the random forest model to investigate the effect of the built environment on the duration of light physical activity of older adults. The results showed that streetscape greenery has the most significant impact on older adults’ light physical activity; greenery can be assessed in a hierarchy of areas; population density and land-use mix only have a positive effect on older adults’ light physical activity up to a certain point but a negative effect beyond that point; and a greater distance to the park within 1 km is associated with a longer time spent on light physical activity. Therefore, we conclude that the built environment’s impact is only positive within a specific range. Changes in the intervention of environmental variables can be observed visually by calculating the relative importance of the nonlinearity of built environment elements with partial dependency plots. These results provide a reasonable reference indicator for age-friendly community planning.

## 1. Introduction

China has become an aging society since 2000 [1]. Approximately 264 million older adults (aged 60+ years) accounted for 18.4% of China’s total population in 2021. According to the “China Development Report 2020: Trends and Policies on Population Aging in China”, it is predicted that China’s aging population will account for 30% of the total population in 2050 [2]. Therefore, to cope with the upcoming heavily aging society, there is a need to intervene and improve older adults’ physical health and quality of life by encouraging active travel and physical activity [3]. 

Light physical activity is an essential means of supporting health and well-being [4,5]. It helps older adults maintain autonomy while promoting physical function [6]. Studies have shown that at least 150 min of light physical activity per week for older adults reduces the risk of contracting infectious diseases and developing slow-moving diseases [7]. Light physical activity, such as walking, can help older adults reach their destinations and increase social participation, self-satisfaction, and well-being [8].

Many studies have analyzed the built environment’s impact on light physical activity of older adults. For example, updating the community environment with more compact and multifunctional areas, parks and green spaces, and more aging-specific infrastructure increases the propensity of older adults to engage in light physical activity [9,10,11,12]. However, most current research used various models (e.g., logistic regression models [13], structural equation models [14], negative binomial regression models [15], and difference-in-differences models [16]) to analyze the relationships between the built environment and physical activity. It suggests that the built environment has a nonlinear effect on older adults’ propensity to travel and walk, with relevant environmental variables only trending with travel time and inclination within a particular range, with both too high and too low values leading to a decrease in travel time and propensity [17,18].

Theoretically, the nonlinear and threshold impacts of the built environment on light physical activity may be influenced by two factors [19,20]. The first relates to peer effects or collective socialization that develops among community exercisers [21]; the second relates to the development of the impact of physical activity. In most regional studies, urban density positively affects light activity; however, in Hong Kong, increasing urban density decreases the longevity of light physical activity [10,22]. In this context, there is a need for a more detailed understanding of the built environment’s impact on light physical activity among older adults. Ignoring the nonlinear relationship may lead to misguided policy interventions by the government [23]. Therefore, it is necessary to sort out and account for the potential nonlinear effects of the built environment on older adults’ light physical activity.

This present study has multiple contributions. Firstly, examining the nonlinear and threshold impacts of the built environment on light physical activity among older individuals validates and complements previous research. Secondly, by revealing well-established relationships, the results show subtle effects to effectively provide appropriate environmental interventions and policy actions to encourage light physical activity among older adults. Thirdly, the methodology uses the random forest model to capture the complex relationships between time and environmental variables for light physical activity. Deep learning is also used to compute urban street scenes to more objectively reflect the local environment in which senior citizens reside.

## 2. Literature Review

### 2.1. Built Environment and Light Physical Activity among Older Adults

The literature from 2005 onwards was searched using the Web of Science for relevant keywords to ensure that relevant theories were current. Walking and jogging are the most prevalent kinds of light physical activity among older people, with significant health and autonomy advantages, and the impacts of the built environment on light physical activity among elderly individuals have been widely established in the literature [24,25,26,27,28]. In some cross-sectional studies, positive effects have been shown between light physical activity and various environments, including population density, land-use mix, street connectivity, and distance to destinations (e.g., public transport stops and park/plaza destinations) [12,29].

In these studies, community population density was most often analyzed and was found to be associated with a tendency to walk as the primary mode of transport [17,30,31]. A higher land-use mix is also positively associated with older adults’ propensity for light physical activity [14,32]. Some studies have shown that the propensity and duration of light physical activity among older adults increase when recreational facilities such as parks, squares, or green spaces are available near the community [33,34,35,36]. The effects of the built environment on light physical activity among older adults are stronger than those on younger adults [37,38]. Furthermore, the various built environment factors that influence light physical activity among older adults may interact rather than act independently [14].

Streetscape greenery has recently attracted scholarly attention as an important feature of the built environment [34,39,40,41,42]. This is largely due to advances in machine learning upgrading streetscape greenness measurement techniques, the ability to accurately and efficiently estimate perceived greenness at eye level by computer, and the availability of high-definition and more current streetscape image data that can be purchased for a fee from mapping websites. In addition, less attention has been paid to this nonlinear effect of the built environment on light physical activity among older adults [17,43].

### 2.2. Nonlinear Relationships between Light Physical Activity and the Built Environment

Recent research has uncovered nonlinear and threshold interactions between mild physical activity and the built environment [17,23,44,45], although empirical research on nonlinearity implies that the marginal influence of the explanatory variable on the result is dependent on the variable’s value, which is not always the case [46]. Threshold effects are important in nonlinear relationships, where the change in impact increases or decreases beyond a threshold, and partial dependence plots can be a good illustration of this effect [47]. The built environment may have different effects across the entire sphere of influence; hence, finding the right sphere of influence is more economically efficient for policy implementation [48].

It has been suggested that the built environment has a nonlinear effect on light physical activity among older adults [10]. It was found that a more excellent land-use mix in highly dense areas of Hong Kong did not increase or decrease the propensity for older adults to engage in light physical activity. Researchers have recognized the intricacy of the built environment’s influence on the walking of older persons and indicated the potential advantages of examining nonlinear and threshold effects to develop more effective environmental treatments [19,24,49,50]. In the most recent study on light physical activity among older adults, land-use mix thresholds showed an increasing trend between 0.40 and 0.65 and a decreasing trend beyond 0.65 [17]; another study showed that the highest threshold for land-use mix showed an increasing trend from 0.20 to 0.55 and a decreasing trend from 0.55 to 0.80 [18]. This suggests that nonlinear effects of environmental factors vary across regions. However, very little research has been conducted on the nonlinear relationships between the built environment and mild physical activity among older adults; thus, there is a need to continue to add validated data results from different regions to effectively and scientifically capture changes in environmental factor thresholds and ensure effective resource-saving in a social context.

## 3. Data and Methodology

### 3.1. Light Physical Activity Data

To continue adding to the research on the nonlinear relationship between the built environment and light physical activity among older adults, this research was based on the 2021 medical examination reports of older adults from community health service centers in Lanzhou. The results included self-reported weekly hours of light physical activity among older adults in the communities. Lanzhou, the capital city of Gansu Province, China, is a rather representative case, as the elderly population has been increasing in recent years; in 2020, older adults accounted for approximately 11.7% of the city’s total population. With the decline in the younger population, the effects of aging have become more severe.

With the support of the Gansu Provincial Health Commission, Lanzhou City is required to conduct medical examinations for older adults in each community every 6 months to summarize their health status, thus facilitating management for timely policy adjustments, including individual sociodemographic information and the need for each person to self-report their light physical activity time. Data in this study were taken from March to June 2021, avoiding the cold and hot seasons to ensure more exercise probability. Of the eight districts in Lanzhou, Chengguan and Qilihe districts had the largest populations and the highest proportion of older adults. In addition, a total of 14 sample areas with high SES (socioeconomic status) and low SES were selected on the basis of three different build-up densities (see Figure 1), with a total of 1773 valid data after removing missing self-reported activity time (see Table 1).

### 3.2. Environmental Variables

The choice of predictor variables was guided by the empirical nature of the literature and data, constructed into an environmental assessment framework through the “5Ds”. The “5Ds” are the five components of the built environment that are important in terms of their impact: density, design, diversity, distance to destination, and distance to transport [51]. We had a total of two sociodemographic variables (age and gender) and 10 environmental variables (with population density, land-use mix, street connectivity, intersection density, distance to bus stops, number of bus stops, distance to parks, number of parks, number of street crossings, and streetscape green views) as independent variables. All environmental variables were calculated within the ArcGIS software framework, with environmental data provided by the online mapping service Gaode Maps (https://en.tongdajiaju.cn/maps.html, accessed on 10 March 2022). Table 2 gives description and summary statistics of the predictor variables, and the average weekly duration of light physical activity among older adults was 84.83 min.

**Table 2 ijerph-19-08848-t002:** Descriptiveon and summary statistics of the predicted and predictor factors.

Variable	Description	Mean	SD
Predicted variable (dependent variable)			
Light physical activity	Weekly duration of light physical activity for older adults (unit: min).	84.83	55.03
Predictor variables: sociodemographics (independent variable)			
Age	Older adults aged 60–69 = 1, older adults aged 70–79 = 2, older adults aged ≥80 = 3	1.70	0.74
Gender	Male = 1, female = 2	1.58	0.49
Predictor variables: built environment (independent variable)			
Population density	The neighborhood’s population density (unit: 100 persons per km^2^)	0.71	0.01
Land-use density	Entropy for local land uses $H=-\left[\overset{N}{\underset{I=1}{\sum }}P_{i}\times \ln\left(P_{i}\right)\right]/\ln\left(N\right)$, where $P_{i}$ represents the percentage of the *i*-th land use, and *N* represents the total number of land-use categories. Seven land uses are studied (*N* = 7): residential, office, commercial, medical, entertainment, public services, and education	0.65	0.12
Street connectivity	Total sidewalk length/total built-up area in a buffer zone (km/km^2^)	1.95	0.39
Road intersection density	Within-community density at a street intersection (unit: 1 km^2^)	26.26	6.45
Number of bus stops	The total number of bus stops inside a 1 km buffer zone.	33.71	9.47
Bus stop distance	The shortest distance from the sample plot to the bus stop	124.14	110.60
Number of parks	The total number of groups inside a 1 km buffer zone.	1.41	0.94
park distance	The shortest distance from the sample plot to the park	319.24	266.53
Number of overpasses	The total number of overpasses inside a 1 km buffer zone.	3.09	1.74
Streetscape greenery	Sampling points generated by taking a fixed 50 m spacing for all streets within the buffer zone, based on the zoning of the sampled elderly area. (static maps were purchased from the Baidu Maps developer platform, and a total of 29,000 BSV images were collected and purchased; for each location point, four images were sampled at 90°, 180°, 270°, and 360° to represent a 360° panoramic image; the Baidu Street View-generated streetscape greenery was calculated as follows [39,52]: *Green view index* = $\overset{4}{\underset{i=1}{\sum }}Greenery\;pixels_{i}/\overset{4}{\underset{i=1}{\sum }}Total\;pixels_{i}$)	0.16	0.02
Sample size	1773		

### 3.3. Computational Methodology

Random forest (random regression forest is used in this study) is one of the most influential machine learning algorithms in the international arena [53]. Compared to traditional linear regression models (a linear regression model is a training exercise on a sample of data to learn a hypothesis function $\;h\left(x^{i}\right)$, such that $h\left(x^{\left(i\right)}\right)=wx^{\left(i\right)}+b\approx y^{\left(i\right)}$ [54]), it contains multiple re-regression trees to reduce the risk of fitting, and it has better ease of interpretation, multi-feature processing, easy-to-extend classification, and scale-free properties [55]. It can perform data mining and classification and regression tasks to explain finer correlations between variables [56].

The random forest approach shown in Figure 2 incorporates a stochastic process, with a small number of differences in each decision tree, to reduce the prediction variance by combining the predictions of individual regression trees, thus optimizing the model incrementally [57]. In a regression tree, the sum of the squared residuals of the regression is the sum of the squared residuals of this tree at the root of the regression tree, after which a variable, also called a feature, is selected that minimizes the sum of the squared residuals of the two branches of the tree. This criterion is used to select subsequent attributes at the two nodes of the bifurcation. This is repeated until a complete tree is produced. The final prediction is a weighted distribution of the target variables at the leaf nodes.

The random forest studies used in this study were all called using the “sklearn” learning library (Scikit-learn is an open-source machine learning library that supports supervised and unsupervised learning, which also provides various tools for model fitting, data preprocessing, model selection, model evaluation, and many other utilities [58]), using the GridSearchCV tool provided by “sklearn” in the model tuning, which helps the model to automatically tune the parameters by systematically traversing multiple parameter combinations and determining the best effect parameters through cross-validation. However, this method is only applicable to small data levels, and the greedy algorithm can be used to obtain a parameter tuning with the greatest impact, followed by bagging to optimize it to achieve the optimal value [59].

In contrast to linear regression, where the former is a linear association between predictors and predictor variables, the random forest model does not make these assumptions. Therefore, models that make assumptions on the basis of predictor variables produce a partial dependence plot (PDP) to account for the link between the test and predictors [59]. The equation for partial dependence is as follows:$$\hat{f_{s}}\left(x_{s}\right)=\frac{1}{N}\sum_{i=1}^{N}\hat{f}\left(x_{s},x_{ic}\right),$$
where $x_{ic}$ is the $x_{c}$ value of the variable in the modeled dataset, and *N* indicates the number of occurrences. The graph depicts the influence of the built environment on walking time in a variety of nonlinear relationships [60].

Three factors can significantly affect the predictive effectiveness of the random forest algorithm [61]: improving the accuracy of individual trees, reducing the similarity of each tree, and parameterizing the entire model (most notably via three parameters [62]: maximum tree depth, total tree count, and the number of splits).

Before modeling, the independent variables were first checked for multicollinearity analysis, with all sociodemographic and environmental variables satisfying VIF (variance inflation factor) < 5, ensuring that all variables were free of multicollinearity. Light physical activity times were normalized for ease of calculation, and, for the optimization of random forest model pairs, the range of these three parameters was first determined (maximum tree depth, 1–30; number of features per tree, 2–10 and 10–1000, with one interval per ten trees). Next, a total of 24,000 (30 × 8 × 100) potential combinations were estimated, and the square-root MES was used to evaluate model performance [63]. After 24,000 tests, the model stopped developing at a maximum tree depth of 6, a feature count of 2, and a tree count of 800. The final model functioned well. Subsequently, the model was applied for further analysis.

## 4. Results

### 4.1. Relative Importance of Predictor Variables

Table 3 shows the relative relevance of the predictors, which are ranked in Figure 3. Other “5Ds” environmental factors were ranked after streetscape greenery according to overall relative importance, with streetscape greenery having a relative importance of 12.48%. Notably, population density was ranked relatively low, being in ninth place with a value of 4.98%, demonstrating that streetscape greenery as an environmental variable has a greater impact on light physical activity than the traditional “5Ds”, which indicates that greenery attracts the attention of older adults more than other environmental elements [64].

Environmental variables accounted for 65.36% of the total importance, while sociodemographic variables accounted for 34.64%. This suggests that built environment characteristics significantly influence older adults’ light physical activity, a finding consistent with Yang and Cheng et al. [17,18], who examined the relationship between walking time and walking propensity and the built environment using older adults in Hong Kong, China, and Nanjing, Jiangsu Province, China. The nonlinear relationship between walking time and built environment was investigated using a random forest nonlinear model, with built environment factors having a significant effect and being more than 50% important in both models. In previous nonlinear studies in Europe and North America, the meaningful relationship between sociodemographic and built environment variables was less transparent [65,66,67], and this study demonstrates that created environment variables have a greater impact on light physical activity than sociodemographic variables.

Figure 3 shows that the traditional “5Ds” were in the top positions in terms of importance except for population density, with the highest being road intersection density (9.17% importance), followed by distance to parks in fifth place (8.10% importance), land-use mix in sixth place (7.66% importance), distance to bus stops in seventh place (6.93% importance), street connectivity in eighth place (6.00% importance), and population density in ninth place (4.98% importance). The lowest importance of the number of footbridges and destinations for the impact of light physical activity on older adults may be due to the relatively low impact of park resources and footbridge facilities, as they are relatively homogenous, and local older adults do not have more options for light physical activity. Abundant bus stops attract older adults to walk for and use bus transport.

### 4.2. Nonlinear Effects of Environmental Variables

Figure 4a–j show the partial dependence plots for the built environment variables, with the black line showing the change in the effect of environmental variables on light physical activity and the red line showing the fitted curve for the change in the effect of environmental variables; the fitted curve provides a more visual response to the trend of change. Figure 4a shows that the predictive index of population density was positively associated with older people and light physical activity at 0.065–0.075, a result that is consistent with existing studies [17,18]. This may be due to a lack of security among older people at this population density between 0.045 and 0.65, followed by an increase in physical activity due to older people’s preference for a lively community atmosphere and a safe place to be physically active [68].

The nonlinear effect of land-use mix on light physical activity can be seen in Figure 4b, with a positive effect when the predicted index of land-use mix was between 0.20 and 0.65, a result that is consistent with existing research [12,17], with a peak of 0.65 indicating that such a land-use mix provides a more prosperous functional need and an enhancement for older adults performing light physical activity. A predicted index of 0.65–0.80 is associated with a decreasing trend in light physical activity, demonstrating that a high land-use mix also negatively affects light physical activity [29].

Figure 4c expresses the effect of street connectivity on light physical activity in older people, with an overall downward trend in the curve; the best effect of light physical activity can be seen at a predicted value of 1.60 with a negative effect above 1.60, which can be attributed to dense and complex street connections causing disorientation among older adults [24,50], as well as a decrease in safety and perception of older adults crossing the road, along with the reduced physical activity. Therefore, controlling the number of road crossings while considering the planning of additional road crossings is essential for light physical activity. Figure 4d shows the nonlinear effect of road intersection density on light physical activity. The curve shows a tendency for the predicted values to increase between 10 and 33 and plateaus after reaching 33, which is consistent with the positive association between road intersection density on light physical activity [26].

Figure 4e,f show the nonlinear effects of the number of bus stops and the shortest distance to the bus stop on light physical activity among older adults. Figure 4e reveals that the number of bus stops increases with a slight rise and fall at 30, while the predicted value peaks at 10–15, in line with the findings of Cheng et al. [17]. In previous studies, the number of bus stops was positively associated with light physical activity [17,69]; however, in this study, a decreasing trend was found when there were more than 15 bus stops associated with light physical activity, which may also have been due to the number of buses stops covering a dense area. The predicted values in Figure 4f peaked between 0 and 100 m, showing a decreasing trend with light physical activity for older adults above 100. This prediction is consistent with the results of Yang.

Figure 4g,h show the nonlinear effects of the number of parks and the shortest distance to the park, with the highest peak at 1 in Figure 4g being negatively associated with light physical activity among older adults, which may be a result of older adults being accustomed to visiting their usual parks and an increase in the number of parks not increasing the attraction for light physical activity among older adults. One point worth noting in Figure 4h is that increasing park distance was positively associated with light physical activity among older adults, unlike most previous studies [17,70], which may be due to locality, where older adults choose to walk as their mode of transport to parks. Figure 4i shows the nonlinear effect of the number of footbridges on the light physical activity of older people, peaking at predicted values between 0 and 2, and trending downward beyond 2. In traditional linear studies, the number of footbridges was negatively associated with physical activity among older adults. Nevertheless, in this study, the effect on light physical activity was positive when the number of footbridges was between 0 and 2, which may be due to road restrictions on pedestrian access. This may be due to the fact that roads restrict pedestrians from having to cross footbridges, thus increasing the opportunity and time for light physical activity. Above a value of 2, the road makes it more difficult for older adults to walk and discourages them from light physical activity.

Figure 4j shows the nonlinear effect of streetscape greenery on light physical activity among older adults as the first of the environmental variables, with a peak at predicted values of 0.12–0.15, a trend consistent with Yang’s study [18,49], and a relatively limited effect of greenery, with a decreasing trend beyond 0.15. This result may be due to the fact that Lanzhou is located in northwest China, where the relatively low greening rate does not allow for the observation of a higher greening rate, but it is expected that a greening rate in the range of 0.12–0.15 can achieve a good promotion of light physical activity in a low greening city.

### 4.3. Comparative Analysis of Random Forest and Linear Regression Models

We evaluated the performance of random forest and binary logic modeling using a tenfold cross-validation approach. We used two commonly used classification metrics: model accuracy and mean squared error. These two metrics were calculated as follows:$$MAE=\frac{1}{N}\overset{N}{\underset{i=1}{\sum }}{\left(\hat{y_{i}}-y_{i}\right)}^{2},$$
$$RMSE=\sqrt{\frac{1}{N}\overset{N}{\underset{i=1}{\sum }}{\left(\hat{y_{i}}-y_{i}\right)}^{2}},$$
where N is the total number of respondents in the validation set, $\hat{y_{i}}$ is the predicted walking propensity of the sample, and $y_{i}$ is the actual walking propensity; lower values indicate a more accurate model. Table 4 shows the results of the analysis of the Wilcoxon rank sum test for 10 cross-validations of the two models. It can be seen that the paired medians of MAE (Z = 2.805 **, *p* = 0.005) and RMSE (Z = 2.803 **, *p* = 0.005) for the random forest model were both smaller than the values for the linear model; thus, the random forest model outperformed the linear regression equation and was superior in terms of nonlinear performance.

## 5. Discussion

In order to create an age-friendly community, a community environment that offers light physical activity facilities allows older adults to engage in preferred outdoor activities and maintain physical autonomy. A good understanding of the impact of the built environment on older adults’ physical activity is important and has profound implications for urban and transport planning [71]. In this study, we elucidated the nonlinear association between the built environment and the duration of light physical activity among older adults. The nonlinear relationship was shown to be universal, with different built environment attributes having different nonlinear effects. A random forest modeling approach was used. Compared to traditional linear regression methods, the random forest approach was able to reveal the complex relationship between physical activity and the built environment more effectively through relative importance and, in part, dependency plots. This provides validation for previous studies and support for subsequent studies.

The nonlinear effects of the built environment on light physical activity of older adults were found to have an effective range of 0.65–0.75 for 100,000 people/km^2^ for population density and 0.20–0.65 for land-use mix, with too high a population density and too high a land-use mix having a negative effect on light physical activity among older adults. This finding is the same as the study by Lu and Chen et al. [17,72]. The effective ranges for street connectivity and road intersection density are 1.6 and 10–33 km/km^2^, and the impact of nonlinearity needs to be considered in street design, with road intersections not being too dense and with safety and orientation cues. Light physical activity is promoted among the elderly when traveling on public transport, and there is no significant impact on light physical activity when there are more than 10 bus stops. The more distant a park is within 1 km, the more it promotes light physical activity time for older adults, and the more distant a park is, the more it can be planned to enhance activity time for older adults. The predicted index of streetscape greenery is 0.12–0.15, which has a positive effect on light physical activity and can be graded in terms of the nonlinear effect of the streetscape, with a greenery rate of 0.12–0.15 in drought-prone areas satisfying the need for light physical activity for older adults [66,73]. Overall, to achieve the enabling impacts of the built environment, subtle environmental interventions are necessary.

The relative importance of the built environment relative to sociodemographic data is overwhelming. Moreover, streetscape greenness can effectively influence light physical activity among older adults [18,49]. This is followed by the design of the street and then the distance to the park, which significantly impacts the duration of light physical activity among older adults. Therefore, urban streetscape greenery should be a primary consideration. Controlling the distance of bus stops from residential areas and the mix of land uses is also closely related to light physical activity for older adults. These indicators can be taken into account when designing communities in terms of light physical activity for older adults.

## 6. Conclusions

There were limitations to this study. Firstly, most prior research focused on first-tier developed cities in China. We provided a case study based on Lanzhou, an industrialized, inland city in northwest China with a moderate population and urban density. The applicability of the findings to other cities is uncertain. Secondly, in actual walking, older adults can experience a richer experience through greenery; as not all greenery captures the perceptions of older adults, while also changing across seasons, long-term longitudinal data collection is required for validation of these effects. This study was also a practical application of a new methodology. The nonlinear and threshold effects revealed can provide important insights for considering land-use and transport policies that encourage walking among older adults.

## Figures and Tables

**Figure 1 ijerph-19-08848-f001:**
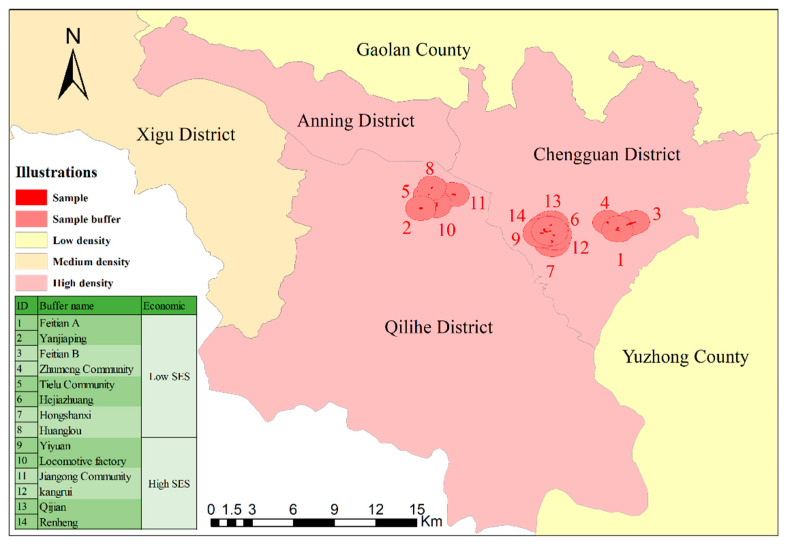
Research sample screening diagram.

**Figure 2 ijerph-19-08848-f002:**
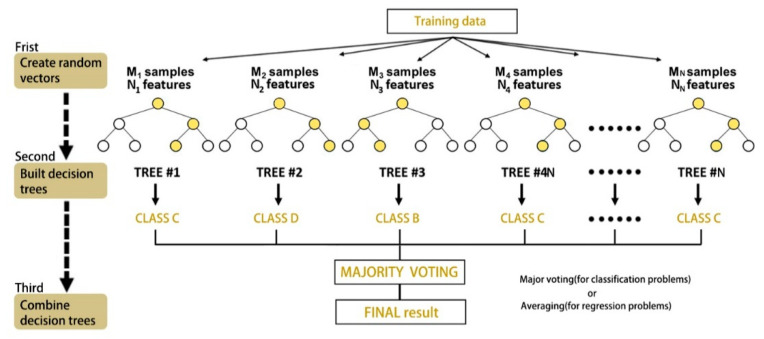
A random forest technique example.

**Figure 3 ijerph-19-08848-f003:**
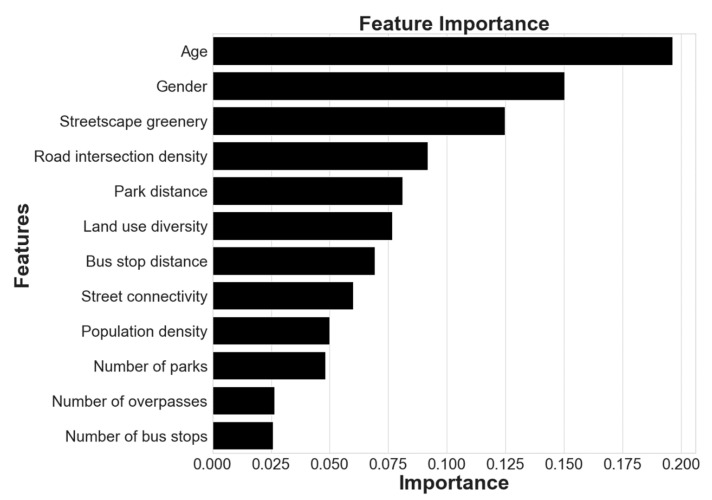
Predictor variables’ relative relevance.

**Figure 4 ijerph-19-08848-f004:**
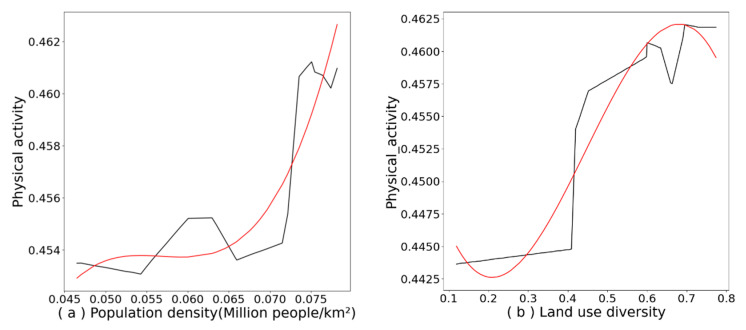
The nonlinear effect of built environment characteristics on physical activity.

**Table 1 ijerph-19-08848-t001:** Research regions.

ID	Buffer	Prices (Million)	Economic	ID	Buffer	Prices (Million)	Economic
1	Renheng	2.00	High SES	8	Hongshanxi	1.30	Low SES
2	Qijian	1.70		9	Hejiazhuang	1.27	
3	Kangru	1.60		10	Zhumeng community	1.20	
4	Yiyuan	1.50		11	Tielu community	1.20	
5	Locomotive factory	1.50		12	Yanjiaping	1.10	
6	Jiangong community	1.50		13	Feitian B	1.10	
7	Huanglou	1.40	Low SES	14	Feitian A	1.00	

Buffer zone house prices greater than or equal to 15,000 USD are considered high SES, while house prices below 15,000 USD are considered low SES.

**Table 3 ijerph-19-08848-t003:** The random forest algorithm calculates the relative relevance of predictor variables.

Category	Variable	Rank	Relative Importance (%)	Total (%)
Sociodemographics				34.64
	Age	1	19.63	
	Gender	2	15.01	
Built environment				65.36
	Population density	9	4.98	
	Land-use density	6	7.66	
	Street connectivity	8	6.00	
	Road intersection density	4	9.17	
	Number of bus stops	12	2.58	
	Bus stop distance	7	6.93	
	Number of parks	10	4.81	
	park distance	5	8.10	
	Number of overpasses	11	2.65	
	Streetscape greenery	3	12.48	
Total relative importance				100

**Table 4 ijerph-19-08848-t004:** Results of Wilcoxon analysis of MAE and RMSE for random forest and linear regression models.

Model	Median (P_25_, P_75_)	Z	*p*
Random forest MAE	0.486 (0.5, 0.5)	2.805	0.005 **
Linear regression model MAE	0.492 (0.5, 0.5)		
Random forest RMSE	0.492 (0.5, 0.5)	2.803	0.005 **
Linear regression model RMSE	0.496 (0.5, 0.5)		

** *p* < 0.01.

## Data Availability

The data are available from the corresponding author upon reasonable request.

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
