# Peer review of "Nonlinear Effects of the Built Environment on Light Physical Activity among Older Adults: The Case of Lanzhou, China"

_ijerph, 2022, doi:10.3390/ijerph19148848_

Round 1
Reviewer 1 Report
1. Abstract: this part should be rewritten to improve the reading. For example, the first Paragraph.
2. Introduction: there are many parts which may need the references.
For example, the first eight lines.
3. Methods:
Section 3.1: the urban neighbourhoods studied should be clearly presented in the Figure 1. It would be better to only keep the ‘Pink area’ and enlarge the parts of neighbourhoods studied.
In this part, a table with details of social-economic information could be applied to enhance the understanding of these neighbourhoods.
Section 3.2:
This part should be rewritten to give a clear explanation of various variables.
Table 1 should be modified to show predictors and dependent variables in a more clear way.
Section 3.3:
This part should be revised to reduce the theoretical explanation of statistical model. But, the application of this model in your study should be enhanced.
Which statistical software / platform was applied in terms of Random forest approach? R?
The footnote 1 should be replaced by references.
Here, the Figure ‘3’ should be ‘2’.
4. Results:
Section 4:
Section 4.2 & Figure 4: you may need to explain the significance range of non-linear effects in terms of the Random Forest. It would be required to marked in the curves.
Section 4.3: the binary logical regression should be explained in the part of Method.
5. Discussions and conclusions
Section 5:
For the discussions, you may need to apply references (similar research) to help explain your results.
6. This paper presented an interesting study. However, the English writing and the application of terms have some improper parts. Please have a proof-reading to enhance the reading.
1. Abstract: this part should be rewritten to improve the reading. For example, the first Paragraph.
2. Introduction: there are many parts which may need the references.
For example, the first eight lines.
3. Methods:
Section 3.1: the urban neighbourhoods studied should be clearly presented in the Figure 1. It would be better to only keep the ‘Pink area’ and enlarge the parts of neighbourhoods studied.
In this part, a table with details of social-economic information could be applied to enhance the understanding of these neighbourhoods.
Section 3.2:
This part should be rewritten to give a clear explanation of various variables.
Table 1 should be modified to show predictors and dependent variables in a more clear way.
Section 3.3:
This part should be revised to reduce the theoretical explanation of statistical model. But, the application of this model in your study should be enhanced.
Which statistical software / platform was applied in terms of Random forest approach? R?
The footnote 1 should be replaced by references.
Here, the Figure ‘3’ should be ‘2’.
4. Results:
Section 4:
Section 4.2 & Figure 4: you may need to explain the significance range of non-linear effects in terms of the Random Forest. It would be required to marked in the curves.
Section 4.3: the binary logical regression should be explained in the part of Method.
5. Discussions and conclusions
Section 5:
For the discussions, you may need to apply references (similar research) to help explain your results.
6. This paper presented an interesting study. However, the English writing and the application of terms have some improper parts. Please have a throughout proof-reading to enhance the reading.
Author Response
Response to Reviewer 1 Comments
Point 1: Abstract: this part should be rewritten to improve the reading. For example, the first Paragraph.
Response 1: This advice is very valuable. This section of the abstract was freshened up and the first paragraph was rewritten (see page 1).
Point 2: Introduction: there are many parts which may need the references.
Response 2: Thank you for your advice, References have been added to this section of the introduction to ensure that the relevant background can be checked (see page 1).
Point 3:Section 3.1: the urban neighbourhoods studied should be clearly presented in the Figure 1. It would be better to only keep the ‘Pink area’ and enlarge the parts of neighbourhoods studied.
Response 3: Thank you for your suggestion. The urban neighbourhoods studied should be in Figure 1. Only the "pink area" has been retained in the expanded part of the studied neighbourhood (see page 3).
Point 4: Section 3.1: In this part, a table with details of social-economic information could be applied to enhance the understanding of these neighbourhoods.
Response 4: Thank you for your suggestion. This advice is very valuable. A table with details of socio-economic information has been added to enhance the understanding of these neighbourhoods (see Table 1).
Point 5: Section 3.2:This part should be rewritten to give a clear explanation of various variables.Table 1 should be modified to show predictors and dependent variables in a more clear way.
Response 5: Thank you for your advice,in Section 3.2 the previous vague formulation was rewritten and the name of each independent variable and a description of the dependent and independent variables was added in Table 2 to ensure a better understanding (see Table 2).
Point 6: Section 3.3: This part should be revised to reduce the theoretical explanation of statistical model. But, the application of this model in your study should be enhanced. Which statistical software / platform was applied in terms of Random forest approach? R? The footnote 1 should be replaced by references. Here, the Figure ‘3’ should be ‘2’.
Response 6: This advice is very valuable. This part was revised to reduce the theoretical explanation of the statistical model. A platform for the use of random forests was added, as well as a description of how the model was applied in this study.(see page 5) Reference indexing error in footnote 1 was corrected and all figure and table names are ensured that they correspond to the text.(see page 5)
Point 7:Section 4.2 & Figure 4: you may need to explain the significance range of non-linear effects in terms of the Random Forest. It would be required to marked in the curves.
Response 7: Thank you for your advice. Fitting curves were added to account for the range of significance of the non-linear effects of the random forest squared partial dependence plot amount, and fitting 1 curves allowed for more intuitive observation of changes in the effects of environmental variables on light physical exercise (see page 7-9).
Point 8:Section 4.3: the binary logical regression should be explained in the part of Method.
Response 8: Thank you for your suggestion. A description of linear regression has been added to the methods part (see page 4).
Point 9:Section 5: For the discussions, you may need to apply references (similar research) to help explain your results
Response 9: Thank you for your suggestion. Section 5: For the Discussion section, some research references have been added to help explain the results of this study (see page 10).
Point 10:This paper presented an interesting study. However, the English writing and the application of terms have some improper parts. Please have a proof-reading to enhance the reading.
Response 10: Thank you for the advice. We got a professional language editor to touch up the article.

Reviewer 2 Report
This paper presents a study of the non-linear effects of some features of the urban environment on older adults' self-reported amount of time spent on light physical activity. The authors used random forest regression as a novel method to model these relationships. They found it able to identify the non-linear effects of said urban features and in doing so they corroborated the findings in some previous works. Finally, they used the outcomes of their fitted model to define optimal values for the urban features.
Comments:
1. I found some difficulties reading the paper, there is substantial room for improvement of English language and style. The choice of bold style for the first paragraph in section 2.1 seems to be a typo. Readers could benefit from the inclusion of the expanded forms of the acronyms (e.g. SES, VIF, etc.) the first time they are mentioned.
2. Non-expert readers might benefit from an explicit list of the traditional 5D variables.
3. Researchers interested in replicating the study would benefit from a description of the computational tools used by the authors (machine learning or statistical software packages and the corresponding functions). Analogously, they could find interesting a citation of the particular method used to assess relative importance.
4. Even though the authors focus on reporting the relative importance of the covariates, it could be interesting assessing the point estimates, standard deviation, and statistical significance of their coefficients.
5. The authors state that the random forest model outperforms traditional linear regression models based on the estimates for their MAEs and RMSEs. Looking at the values reported for their point estimates and standard deviation, I wonder whether there are any statistically significant differences in MAE and RMSE between the random forest model and the linear regression model.
Author Response
Response to Reviewer 2 Comments
Point 1: I found some difficulties reading the paper, there is substantial room for improvement of English language and style.
Response 1: Thank you for the advice. We got a professional language editor to touch up the article.
Point 2:The choice of bold style for the first paragraph in section 2.1 seems to be a typo. Readers could benefit from the inclusion of the expanded forms of the acronyms (e.g. SES, VIF, etc.) the first time they are mentioned.
Response 2: Thank you for the advice. The error in bold has been corrected and the full names have been added to the SES and VIF abbreviations (see page 2).
Point 3:Non-expert readers might benefit from an explicit list of the traditional 5D variables.
Response 3: Thank you for the advice. An explanation of the 5Ds has been added to make it easier for the non-specialist reader to understand (see page 1 ).
Point 4: Researchers interested in replicating the study would benefit from a description of the computational tools used by the authors (machine learning or statistical software packages and the corresponding functions). Analogously, they could find interesting a citation of the particular method used to assess relative importance.
Response 4: Thank you for the advice. Supplemented with machine learning or statistical software packages and corresponding functional descriptions for other researchers to conduct related research (see page 5).
Point 5: Even though the authors focus on reporting the relative importance of the covariates, it could be interesting assessing the point estimates, standard deviation, and statistical significance of their coefficients.
Response 5: This advice is very valuable. Random Forests calculate partial dependency plots via regression trees where the relevant standard deviation is not calculated, but the amount of explanation of the partial dependency plots can be increased by adding fitted curves. Partial dependency plots with partial dependency plots are used in the new revision (see page 8 ).
Point 6: The authors state that the random forest model outperforms traditional linear regression models based on the estimates for their MAEs and RMSEs. Looking at the values reported for their point estimates and standard deviation, I wonder whether there are any statistically significant differences in MAE and RMSE between the random forest model and the linear regression model.
Response 6: Thank you for your suggestion, I rechecked my model and code and found that in the random forest the mean difference is the parent of the calculation, while the squared difference of the linear model calculates the sample, I have unified the squared difference of the two models and the MAE, RASE and squared difference of the random forest are smaller than the linear regression model, Therefore, the random forest model outperformed the linear model in the dataset of this study (see page 10 ).

Round 2
Reviewer 1 Report
My comments have been addressed.
Author Response
Thank you for your review.
Reviewer 2 Report
1. There is still room for improvement of English language and style.
2. The request for a study of the statistical differences in MAE and RMSE between the random forest model and the linear model has not been fulfilled. The authors conclude that the random forest model outperforms the linear model based on the values of the MAE and RMSE point estimations form a 10-fold cross-validation. However, their variances could render the small differences across models not statistically significant; which will not support the said conclusion. A two-sample t-test, a non-parametric Welch's t-test in case of unequal variances, or a non-parametric Wilcoxon rank sum test in case of non-normality could suffice.
Author Response
Point 1: There is still room for improvement of English language and style.
Response 1: Thank you for your suggestion. This revision has changed incorrect English terms and improved the English style.
Point 2: The request for a study of the statistical differences in MAE and RMSE between the random forest model and the linear model has not been fulfilled. The authors conclude that the random forest model outperforms the linear model based on the values of the MAE and RMSE point estimations form a 10-fold cross-validation. However, their variances could render the small differences across models not statistically significant; which will not support the said conclusion. A two-sample t-test, a non-parametric Welch's t-test in case of unequal variances, or a non-parametric Wilcoxon rank sum test in case of non-normality could suffice.
Response 2: Thank you for your suggestion, we used 10 each of the MAE and RMSE output indicators from the random forest and traditional linear models and analysed the differences using the Wilcoxon rank sum test, which showed significant differences between the two models.(see page 11)
